# RecA filament sliding on DNA facilitates homology search

**Kaushik Ragunathan[1†a], Cheng Liu[2†b], Taekjip Ha[1,2,3]\***

[1]Departments of Biophysics; [2]Physics; [3]Computational Biology, University of Illinois, Urbana, United States

**Abstract** During homologous recombination, RecA forms a helical filament on a single stranded (ss) DNA that searches for a homologous double stranded (ds) DNA and catalyzes the exchange of complementary base pairs to form a new heteroduplex. Using single molecule fluorescence imaging tools with high spatiotemporal resolution we characterized the encounter complex between the RecA filament and dsDNA. We present evidence in support of the 'sliding model' wherein a RecA filament diffuses along a dsDNA track. We further show that homology can be detected during sliding. Sliding occurs with a diffusion coefficient of approximately 8000 bp$^2$/s allowing the filament to sample several hundred base pairs before dissociation. Modeling suggests that sliding can accelerate homology search by as much as 200 fold. Homology recognition can occur for as few as 6 nt of complementary basepairs with the recognition efficiency increasing for higher complementarity. Our data represents the first example of a DNA bound multi-protein complex which can slide along another DNA to facilitate target search.

**\*For correspondence:** tjha@illinois.edu

[†]**Present address:** [a]Department of Cell Biology, Harvard Medical School, Boston, United States; [b]Center for Research Computing, University of Notre Dame, Notre Dame, United States

**Competing interests:** The authors have declared that no competing interests exist

**Reviewing editor**: Xiaowei Zhuang, Harvard University, United States

## Introduction

The ubiquitous presence of DNA damaging agents poses a constant threat to genome integrity and protein machineries are required to repair DNA damage. Homologous recombination is one of the pathways involved in double strand break repair (*Cox et al., 2000*). An important step in homologous recombination is the reciprocal exchange of basepairs between complementary DNA molecules during a reaction called strand exchange, which is catalyzed by RecA in *E. coli*. Homologs of RecA, Rad51 and Dmc1 in eukaryotes and RadA in archaea, carry out similar functions during DNA repair underscoring the central role of proteins catalyzing strand exchange across all forms of life (*Bianco et al., 1998*).

The strand exchange reaction involves three steps: (1) pre-synapsis, (2) synapsis, and (3) heteroduplex extension via branch migration. Pre-synapsis involves the assembly of RecA monomers on single stranded (ss) DNA in the presence of ATP with a stoichiometry of 3 nt per monomer (*Di Capua et al., 1982*; *Dombroski et al., 1983*). RecA forms a filament which stretches the ssDNA to a length of 1.5 times the length of B-form DNA (*Stasiak et al., 1981*; *Dunn et al., 1982*; *Stasiak and Egelman, 1986*). During synapsis, the RecA filament finds a homologous double stranded (ds) DNA and catalyzes the exchange of complementary base pairs to form a new heteroduplex product. Subsequently, branch migration mediates the extension of the heteroduplex product (*Cox and Lehman, 1981*).

Homology search is the first step in synapsis and is arguably the most mysterious aspect of the strand exchange reaction (*Barzel and Kupiec, 2008*). A RecA filament has to rapidly and accurately find a homologous sequence in the presence of a vast excess of non-homologous dsDNA. The multivalent nature of the RecA filament enables it to sustain contact with a long dsDNA while allowing for frequent dissociation events that permits rapid sampling of different segments of the DNA, thus facilitating homology search (*Forget and Kowalczykowski, 2012*). Experiments involving the mechanical manipulation of the incoming dsDNA upon binding to the RecA filament identified structural

**eLife digest** The DNA molecules in cells are continuously bombarded with radiation, chemicals and other agents, and it is important for cells to repair the damage caused by these before the process of cell division begins. Most DNA molecules consist of two single strands of DNA that are held together by hydrogen bonds in the familiar double-helix structure. Of the various types of damage that DNA molecules are prone to, double-strand breaks are among the most dangerous because they can lead to cancer if they are not repaired.

DNA molecules use four bases—adenine, cytosine, guanine, and thymine—to store genetic information. In single-stranded DNA these bases are attached to a backbone made of alternating sugar and phosphate groups. A crucial feature of double-stranded DNA is that the sequences of bases in the two strands are complementary to each other—adenine is always paired with thymine, and cytosine is always paired with guanine. However, the hydrogen bonds that hold the pairs of bases together are quite weak, which means that the two strands of the double helix can be pulled apart quite easily. The ease with which these bonds can be formed and broken is crucial for many genetic processes.

One way to repair a double strand break is to replace the damaged stretch of DNA with an undamaged stretch from another DNA molecule. This process of swapping DNA molecules, which is called strand exchange, is catalyzed by a protein that is able to interact with two DNA molecules at the same time. An important first step within this process is identifying the stretch of DNA that can be used to repair the break.

Ragunathan et al. now report evidence from experiments on *Escherichia coli* that support a model in which the protein catalyst (RecA in the case of *E. coli*) combines with a single strand of DNA to form a filamentous DNA–protein complex (RecA filament) that can then slide along a double-stranded DNA molecule to search for a complementary sequence of base pairs. High-resolution fluorescent imaging reveals that the RecA filament is able to sample several hundred base pairs before the filament dissociates from the DNA and rebinds at a different location. The sliding was largely driven by electrostatic interactions between the RecA filament and the double-stranded DNA, and the filament was capable of identifying matching sequences that contained as few as six matching bases.

Ragunathan et al. estimate that sliding is about two orders of magnitude faster at finding matching sequences compared to mechanisms that do not involve sliding, such as models that rely solely on chance encounters between DNA molecules and the RecA filament. By showing that a DNA–protein complex can slide along another DNA molecule to search for a target, these results could lead to new insights into other systems in which it is necessary for protein-nucleic acid complexes to locate a particular sequence of bases.

intermediates which facilitate homology recognition and provided insight into mechanisms by which RecA can discriminate between homologous and non-homologous DNA sequences (*Danilowicz et al., 2012*; *De Vlaminck et al., 2012*; *Peacock-Villada et al., 2012*). However, random 3D collision between a RecA filament and the dsDNA even when aided by intersegmental transfer is unlikely to result in precise alignment of matching sequences. For this reason, RecA filament sliding on dsDNA, even over a short distance range of ten to hundreds of bp would significantly accelerate the search for homology. Although many proteins are now known to diffuse along the DNA via 1D sliding (*Blainey et al., 2006*; *Wang et al., 2006*; *Gorman et al., 2007*; *Bonnet et al., 2008*; *Liu et al., 2008*; *Roy et al., 2009*; *Zhou et al., 2011*) RecA filament diffusion on DNA has not been observed before and in fact an early study ruled out a role for long range (approximately several kb) 1D sliding during homology search (*Adzuma, 1998*). Here, we utilized the high spatio-temporal resolution of single molecule FRET (*Ha et al., 1996*) to examine the possibility of RecA filament sliding during homology search.

Using two and three color FRET measurements, we show that RecA filament slides along dsDNA, primarily mediated by electrostatic interactions. Furthermore, by using sequences with short stretches of homology, we could monitor repeated events of homology recognition and disengagement in real time without full dissociation of RecA filament from dsDNA. RecA-mediated homology search is the first example of a DNA bound multi protein complex which can slide on another DNA during target

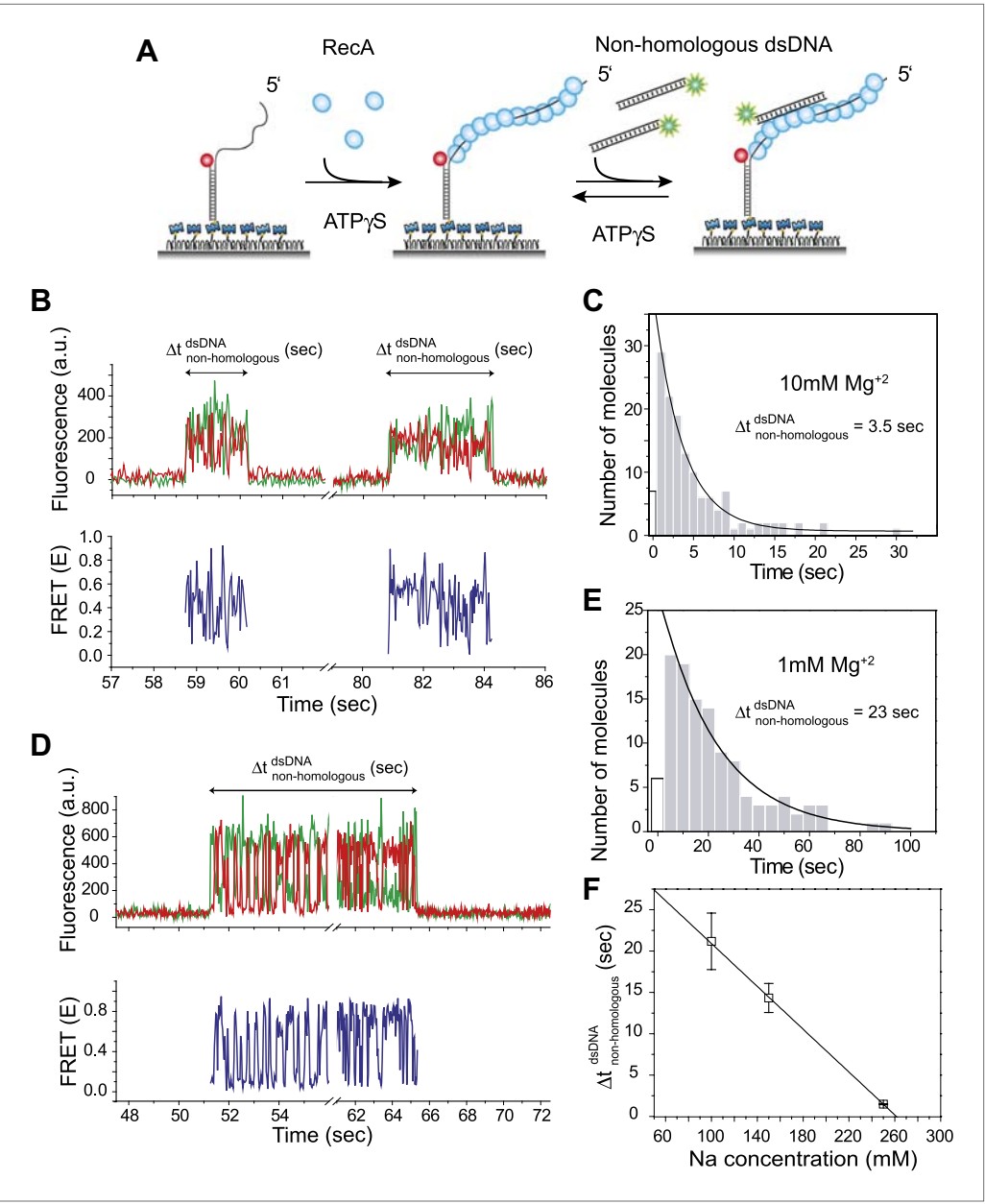

**Figure 1**. Dynamic interactions between RecA filament and non-homologous dsDNA. (**A**) A schematic of the single molecule FRET based assay to detect interactions between RecA filament and non-homologous dsDNA. After RecA filament formation on ssDNA ($L_{filament}$ = 39 nt) labeled with an acceptor (red), a non-homologous dsDNA ($L_{dsDNA}$ = 39 bp) labeled with a donor was added. DNA docking results in appearance of donor (green) signal with FRET reporting on the changes in distance. (**B**) Single molecule time traces showing donor (green) and acceptor (red) intensities exhibits rapid FRET fluctuations with multiple binding and dissociation events within a single time trace (top panel). Corresponding FRET time traces (blue) are shown in the bottom panel (**C**) Histogram of the duration of the bound state for non-homologous dsDNA ($\Delta t_{non-homologous}^{dsDNA}$) and a single exponential decay fit. (**D**) Same as in (**B**), except that the $Mg^{+2}$ concentration in solution was 1 mM. (**E**) Same as in (**C**), with 1 mM $Mg^{+2}$ in solution. (**F**) Plot of $Na^+$ concentration vs dwell time of dsDNA interaction with a RecA filament. $Mg^{+2}$ concentrations in all cases was maintained at 1 mM. Error bars are standard errors of the mean obtained from single exponential decay fitting of dwell times. Linear fitting was used as a guide.

The following figure supplements are available for figure 1.

**Figure supplement 1**. Non-homologous DNA interactions with the RecA filament are independent of ATP hydrolysis.

*Figure 1. Continued on next page*

*Figure 1. Continued*

**Figure supplement 2**. Non-homologous dsDNA interaction with RecA filament: dependence on filament length, $L_{filament}$ (nt) and dsDNA length, $L_{dsDNA}$ (bp).

**Figure supplement 3**. Deletion of acidic residues enhances RecA affinity for non-homologous dsDNA.

search process. It may also serve as a canonical example for other proteins (e.g., telomerase and Argonaute) which are bound to nucleic acid sequences that act as 'guide' strands conferring target site specificity.

## Results

### Dynamic interactions between RecA filament and non-homologous dsDNA

For short (<80 bp) homologous dsDNA substrates, homology search is completed rapidly (within 30 ms) once the dsDNA encounters a RecA filament (*Ragunathan et al., 2011*). Therefore, we used a non-homologous dsDNA to avoid stable product formation and monitor RecA filament in the act of homology search.

We immobilized a partial dsDNA with a 5′ 39 nt ssDNA tail on a passivated quartz surface via biotin–neutravidin interaction (*Figure 1A*). The DNA is labeled with a FRET acceptor (Cy5) at the ssDNA/dsDNA junction. We first formed a stable RecA filament on the DNA by using ATPγS as the cofactor. Then, we added a solution containing non-homologous dsDNA ($L_{dsDNA}$ = 39 bp) and ATPγS while simultaneously removing free RecA from solution, allowing us to observe the interaction solely between the incoming non-homologous dsDNA (free of RecA) and a single isolated RecA filament. RecA filaments formed under these conditions are stable and can carry out the strand exchange reaction with a homologous dsDNA (*Ragunathan et al., 2011*).

Docking of non-homologous dsDNA to the RecA filament is detected as an abrupt appearance of fluorescence signal from the background level. After docking, we observed large and rapid fluctuations in FRET detected as anti-correlated changes of donor and acceptor intensities. The FRET fluctuations are indicative of extensive distance changes between the donor on the dsDNA and the acceptor on the RecA/ssDNA filament. Single molecule time traces show multiple dsDNA binding and dissociation events to the same filament because the observed interactions are transient without forming a stable product (*Figure 1B*). The lifetime of the binding events is exponentially distributed with an average lifetime of 3.5 s (*Figure 1C*). Filaments formed with ATP displayed similar FRET fluctuations upon docking of non-homologous dsDNA and exhibit comparable dissociation times (*Figure 1—figure supplement 1A–C*). Hence, the fluctuations observed here do not require ATP hydrolysis. Using different lengths for the ssDNA tail ($L_{filament}$ = 50 or 99 nt) did not significantly change the lifetime of the encounter complex (*Figure 1—figure supplement 2A*). In contrast, increasing dsDNA length increased the lifetime presumably due to a larger number of contacts between the dsDNA and the RecA filament (*Figure 1—figure supplement 2B*).

We determined the lifetime of the transient encounter complex as a function of magnesium or sodium concentrations. Rapid FRET fluctuations persisted under all the solution conditions tested while the lifetime of the complex decreased with increasing magnesium or sodium concentrations suggesting that the interaction is electrostatic in nature and can be weakened by increased screening (*Figure 1D–F*). Indeed, eliminating negative charge from the RecA C-terminus (*Lusetti et al., 2003*) by deletion of acidic residues resulted in an increased lifetime of dsDNA bound to the RecA filament (*Figure 1—figure supplement 3*).

### FRET fluctuations are due to sliding of RecA filament

We considered two possible explanations for the large FRET fluctuations observed upon docking of dsDNA to the RecA filament. The first involves limited unwinding of the dsDNA (*Bianchi et al., 1985*) by the RecA filament resulting in separation of the duplex ends. Repeated melting and annealing transitions at the labeled end of the dsDNA could in principle result in FRET fluctuations. We tested this possibility using a dsDNA ($L_{dsDNA}$ = 39 bp) labeled at one duplex end (donor and acceptor

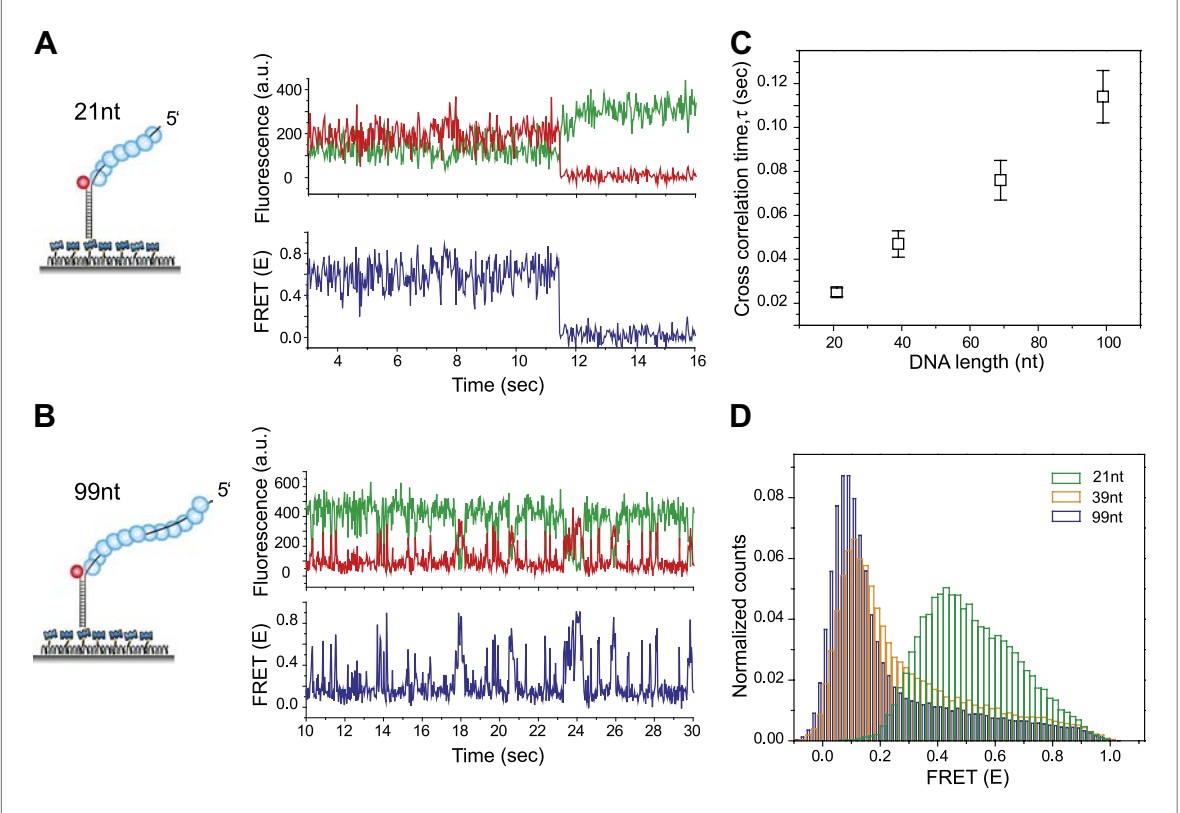

**Figure 2**. RecA filament slides along dsDNA. (**A**) Single molecule traces showing donor (green) and acceptor (red) intensities (top panel) upon docking of non-homologous dsDNA ($L_{dsDNA}$ = 39 bp) to a RecA filament assembled on a ssDNA overhang, $L_{filament}$ = 21 nt. Corresponding FRET time traces (blue) are shown in the bottom panel. (**B**) Same as (**A**), except that the RecA filament is assembled on a ssDNA, $L_{filament}$ = 99 nt. (**C**) Average cross correlation time, vs $L_{filament}$. Error bars are standard errors of the mean determined from three independent datasets. (**D**) FRET efficiency ($E$) histograms of single molecule traces for individual docking events of non-homologous dsDNA to RecA filaments assembled on ssDNA, $L_{filament}$ = 21 nt, 39 nt and 99 nt.

The following figure supplements are available for figure 2.

**Figure supplement 1**. Thermal breathing of DNA ends does not contribute to the observed fluctuations in FRET.

**Figure supplement 2**. Cross correlation and corresponding Monte Carlo simulation of RecA filament sliding.

**Figure supplement 3**. Dependence of sliding rate on solution conditions.

fluorophores on the two opposing strands) so that local melting would cause a FRET decrease and vice versa. When the dsDNA docks to an unlabeled RecA filament immobilized on the surface, we observed stable high FRET (*Figure 2—figure supplement 1*) suggesting that separation of the duplex ends is not the source of FRET fluctuations described in *Figure 1*.

The second explanation involves 1D sliding (or diffusion) of the RecA filament along dsDNA. In order to test the sliding model, we examined whether changes in the length of the RecA filament would affect the time scale of FRET fluctuations since such a change would modulate the encounter frequency between the donor and the acceptor fluorophores. In contrast, conformational changes either within the dsDNA or the protein would presumably exhibit the same time scale of FRET fluctuations independent of RecA filament length.

Shorter RecA filaments ($L_{filament}$ = 21 nt) exhibited more rapid FRET changes and of smaller amplitudes (*Figure 2A*) compared to longer filaments ($L_{filament}$ = 99 nt) which showed larger and slower changes in FRET (*Figure 2B*). To quantify the time scale of FRET fluctuations, we determined cross-correlation of the donor and acceptor fluorescence intensities for four different

filament lengths ($L_{filament}$ = 21, 39, 69 and 99 nt) and a single length of non-homologous dsDNA ($L_{dsDNA}$ = 39 bp). The average cross correlation time increased with increasing filament lengths (*Figure 2C*), supporting the 1D sliding model. Furthermore, the histogram of FRET efficiencies (E) shifted towards lower FRET values for increasing filament lengths (*Figure 2D*). This observation also supports the 1D sliding model because the two fluorophores would spend a smaller fraction of time in close proximity if the filament is longer. To estimate the diffusion coefficient of the 1D sliding process, we performed Monte Carlo simulations of dsDNA diffusing along a RecA filament ('Experimental procedures' and *Figure 2—figure supplement 2*). Given its large persistence length of about 800 nm (*Hegner et al., 1999*), we treated the RecA filament as a rigid rod. We then simulated time traces of donor and acceptor intensities of dsDNA bound to RecA filament of various lengths using different pre-assigned diffusion coefficients ($D_{slide}$) and then calculated the average cross correlation times for each case. By comparing the simulation results with the experimental data we estimated the diffusion constant $D_{slide}$ for 1D sliding of dsDNA relative to the RecA filament to be approximately $0.9 \times 10^{-3}$ µm$^2$/s or 7700 bp$^2$/s. Additionally the time scale of sliding (as measured by the cross correlation time) does not change as a function of sodium or magnesium ion concentrations in solution while the sliding time scale increases twofold when ATPγS is replaced by ATP (*Figure 2—figure supplement 3A–B*). However, the use of ATP as a co-factor, especially in experiments involving short ssDNA substrates is complicated by the dissociation of RecA monomers from the ssDNA ends leading to filament instabilities which could potentially disrupt sliding along the RecA filament track.

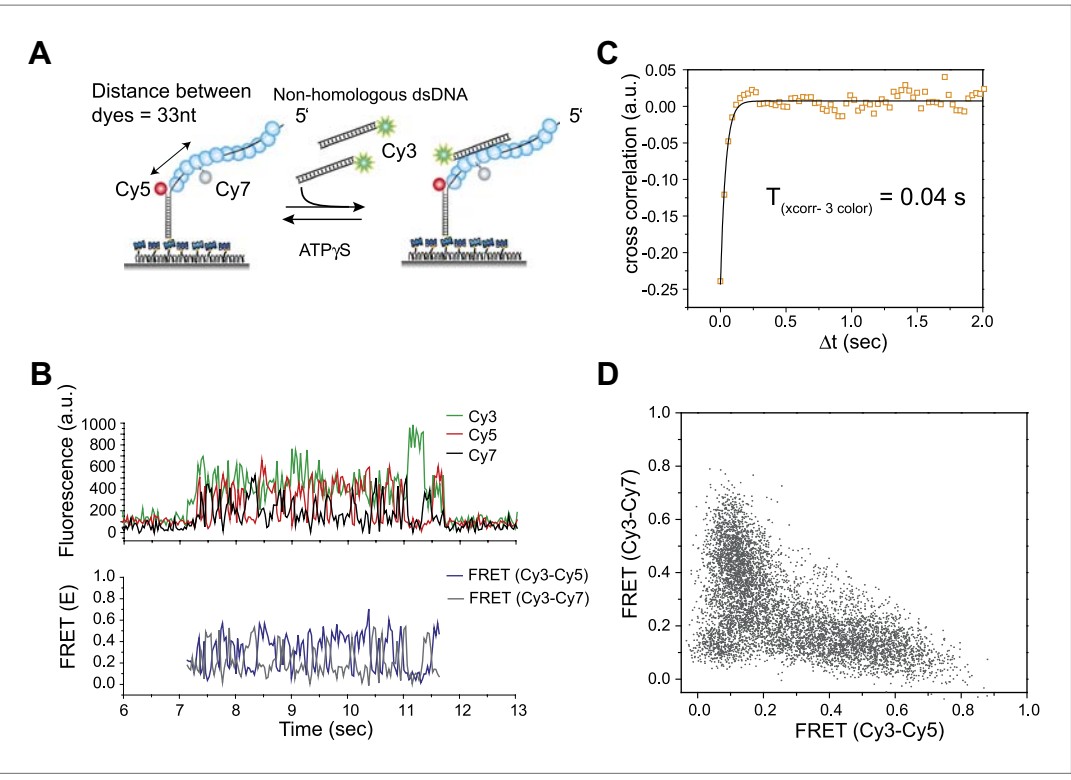

**Figure 3**. Three color FRET observations support RecA filament sliding. (**A**) A schematic of the single molecule three color FRET assay to measure RecA filament sliding. ssDNA ($L_{filament}$ = 99 nt) labeled with two acceptor fluorophores (Cy5-red and Cy7-black) with a separation of 33 nt between the fluorophores, was immobilized on the surface. Upon docking of non-homologous donor (Cy3) labeled dsDNA to the pre-formed RecA filament formation, sliding predicts anticorrelated emissions between the two acceptors. (**B**) Single molecule time traces of Cy3 (green), Cy5 (red) and Cy7 (black) intensities (top panel). Corresponding FRET time traces of FRET between Cy3 and Cy5 ($E_{Cy3-Cy5}$-blue) and FRET between Cy3 and Cy7 ($E_{Cy3-Cy7}$-grey). (**C**) Normalized cross correlation plot of $E_{Cy3-Cy5}$ and $E_{Cy3-Cy7}$ averaged over 30 molecules and a single exponential fit of the data is overlaid (black). (**D**) Scatter plot of $E_{Cy3-Cy5}$ and $E_{Cy3-Cy7}$ for 30 molecules showing unique high FRET regions along both axes.

## Three color experiments support the sliding model

To further test the 1D sliding model, we designed a three-color FRET assay. Here, the immobilized RecA filament is labeled with two different acceptor fluorophores, Cy5 at the ssDNA/dsDNA junction and Cy7 in the middle of the ssDNA embedded within the RecA filament (*Figure 3A*). The two acceptors are separated by 33 nt, which upon RecA binding results in a large separation and negligible FRET between them (*Joo et al., 2006*). The sliding model predicts anticorrelated changes between the two FRET efficiencies, one between Cy3 and Cy5 ($E_{Cy3-Cy5}$) and the other between Cy3 and Cy7 ($E_{Cy3-Cy7}$) because when Cy3 on the dsDNA approaches Cy5, it should move away from Cy7 and vice versa. We observed large and rapid fluctuations in FRET from Cy3 to Cy5 ($E_{Cy3-Cy5}$) and FRET from Cy3 to Cy7 ($E_{Cy3-Cy7}$) (*Figure 3B*, bottom panel). Consistent with the sliding model, the time trace of $E_{Cy3-Cy5}$ and $E_{Cy3-Cy7}$ exhibits anticorrelation between the two FRET efficiencies. The timescale of fluctuations determined from the cross-correlation of the two FRET efficiencies ($T_{xcorr-3color}$ = 0.04 s) (*Figure 3C*) is similar to that measured using the two color assay (*Figure 2C*). The scatter plot of $E_{Cy3-Cy5}$ vs $E_{Cy3-Cy7}$ and calculation of Pearson's correlation coefficient for the two FRET efficiencies ($r_{pearson}$ = −0.6) provide further support for a negative correlation between $E_{Cy3-Cy5}$ and $E_{Cy3-Cy7}$ (*Figure 3D*). Hence, cumulatively the two and three color FRET results support the sliding model.

## Homology detection during sliding

Can the sliding of RecA filament along dsDNA be a physiologically relevant activity, that is, can the RecA filament recognize a homologous sequence during sliding? To answer this question, we embedded two repeats of an identical sequence at positions HS1 and HS2 within an otherwise non-homologous ssDNA (*Figure 4A*). If the target dsDNA contains base pairs which are homologous to the short repeat sequence, it may be possible to observe back and forth sliding events between the two homology sites (HS1 and HS2) without full dissociation of the dsDNA from the RecA filament. Such an observation would indicate that RecA filament sliding allows for homology recognition and base pairing. In all cases, HS1 and HS2 are complementary to a sequence in close proximity to the donor labeled end of the target dsDNA and the location and spacing between the two homology sites were chosen to be within a FRET sensitive regime (approximately 20–80 Å).

We confirmed that dsDNA docking to a RecA filament formed on a ssDNA homopolymer sequence (poly-T sequence, $L_{filament}$ = 50 nt) preserves the large and rapid FRET fluctuations (*Figure 4—figure supplement 1* and *Figure 2B*). We then introduced two identical sequences of length, $L_h$ = 5 nt, at HS1 and HS2 positions which are complementary to a 5 bp sequence within the target dsDNA ($L_{dsDNA}$ = 39 bp). Binding of the dsDNA to the RecA filament formed on the ssDNA with the two 5 nt repeats displayed rapid FRET fluctuations across a broad range of FRET values (*Figure 4B,E*) similar to that observed using a poly-T DNA sequence (*Figure 4—figure supplement 1*). In contrast, when we increased the length of homology at HS1 and HS2 by a single nucleotide to $L_h$ = 6 nt, single molecule time traces displayed transitions between discrete FRET states (*Figure 4C*). The same trend was observed upon further increasing the homology length to $L_h$ = 7 nt (*Figure 4D*). The resulting FRET histograms for 6 and 7 nt homology lengths showed distinct FRET peaks (*Figure 4F,G*).

In order to confirm that the observed FRET peaks arise from base pairing and recognition at specific homology sites, we analyzed different ssDNA sequences containing only one of the two 6 nt homology sites (HS1 or HS2) and obtained a distinct FRET peak at either approximately 0.9 or 0.5, respectively (*Figure 4H,I*). Thus, we assigned the highest FRET state (approximately 0.9) to homology recognition and base pairing at HS1 and the mid FRET state (approximately 0.5) to homology recognition and base pairing at HS2. The lowest FRET state, $E \sim 0.1$, is likely to correspond to dsDNA sliding outside the boundary of HS1 and HS2 in a FRET insensitive regime. Given that this location lacks stable base pairing interactions, we referred to the low FRET state as the non-homologous state (NH). Although the gap between HS1 and HS2 is 5 nt, it is noteworthy that the total distance traversed by dsDNA to exhibit complete basepairing is ($L_h$ + 5) nt. Thus, for the smallest $L_h$ of 6 nt, this translates to a distance of approximately 50 Å (11 nt × 3.4 Å × 1.5). Given the rapid movements of the dsDNA between adjacent homology sites and the fact that both the dsDNA and RecA filament are stiff at the length scales of our experiments, we cannot think of mechanisms other than sliding to explain the observed transitions. We used a statistical approach based on Hidden Markov Model (HMM) analysis to make unbiased assignments of the various FRET states present within each single molecule time trace. The FRET transitions obtained from HMM analysis were then plotted in the form of a transition density plot (TDP) which is a 2D histogram reflecting the frequency

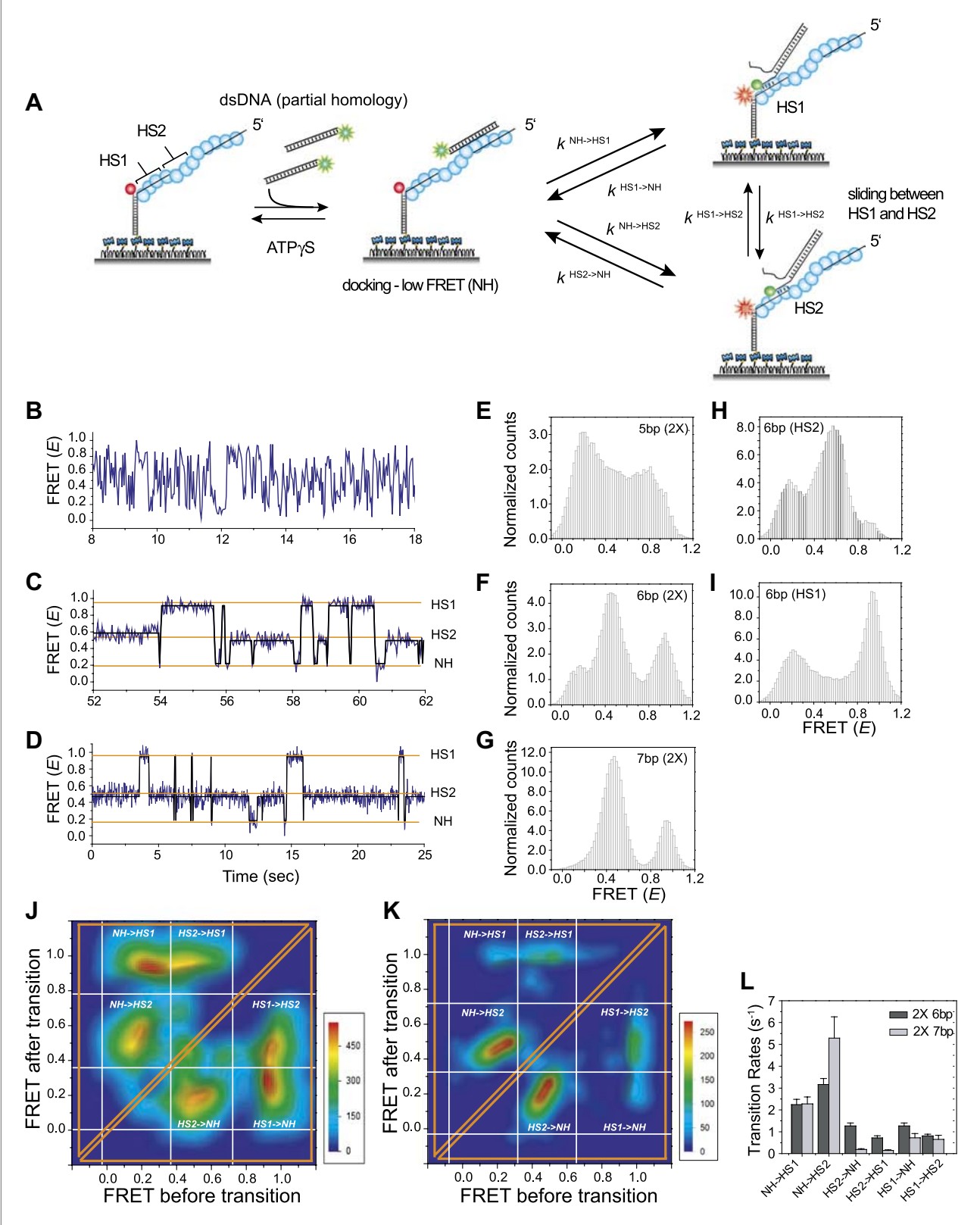

**Figure 4**. Homology recognition during sliding. (**A**) A schematic of the single molecule FRET based assay to detect homology recognition between RecA filament and dsDNA. After RecA filament formation on ssDNA ($L_{filament}$ = 50 nt) labeled with an acceptor (red), a dsDNA ($L_{dsDNA}$ = 39 bp) labeled with a donor was added. Recognition of homology site 1 (HS1) or homology site 2 (HS2) results in the appearance distinct FRET states whose values depend

*Figure 4. Continued on next page*

*Figure 4. Continued*

on their relative distances from the acceptor. Docking at a location along the RecA filament outside a FRET sensitive regime results in low FRET (NH). (**B**) Single molecule time traces showing FRET for an immobilized ssDNA with two identical 5 nt homology sequences at HS1 and HS2 in a poly T sequence background. (**C**) Same as previous, with two identical 6 nt homology sequences at HS1 and HS2 in a poly T sequence background exhibits transitions between distinct FRET states. Idealized time trajectory obtained from HMM analysis is overlaid (black). (**D**) Same as previous, with two identical 7 nt homology sequences at HS1 and HS2 in a poly T sequence background. Idealized time trajectory obtained from HMM analysis is overlaid (black). (**E**) Normalized histograms of single molecule time traces exhibiting FRET for an immobilized ssDNA with two identical 5 nt homology sequences (HS1 and HS2). (**F**) Same as previous, with two identical 6 nt homology sequences (HS1 and HS2). (**G**) Same as previous, with two identical 7 nt homology sequences (HS1 and HS2). (**H**) Same as previous, with a single 6 nt homology site (HS2) at a position distal to the acceptor resulting in the appearance of a distinct mid FRET state. (**I**) Same as previous, with a single 6 nt homology site (HS1) in close proximity to the acceptor resulting in the appearance of a distinct high FRET state. (**J**) Transition density plot (TDP) of all FRET transitions for immobilized ssDNA with two 6 nt repeat sequences (HS1 and HS2) from 236 molecules exhibiting 13,020 transitions. (**K**) Transition density plot (TDP) of all FRET transitions for immobilized ssDNA with two 7 nt repeat sequences (HS1 and HS2) from 191 molecules exhibiting 3,819 transitions. (**L**) Forward and reverse transition rates between HS1, HS2 and NH states for $L_h$ = 6 nt and 7 nt. Error bars denote standard errors of the mean for three measurements.

The following figure supplements are available for figure 4.

**Figure supplement 1**. Non-homologous dsDNA interaction with a poly T ($L_{filament}$ = 50 nt) coated by RecA.

**Figure supplement 2**. Effect of inserting two identical 8 bp homology sites (*HS1* and *HS2*) in a poly T sequence background.

**Figure supplement 3**. Effective rate enhancement of target search due to sliding.

**Figure supplement 4**. dsDNA flipping does not assist RecA mediated homology search.

of transitions between the various FRET states. For $L_h$ = 6 nt, the TDP obtained from 236 molecules exhibiting 13,020 transitions displays three distinct FRET states, $E$ ~ 0.1, 0.5 and 0.9 (***Figure 4J***). Furthermore, the HMM/TDP analysis gave the transition rates and the number of transitions between the different FRET states. The rate of departing HS1 or HS2 was threefold lower than that of leaving NH, showing that even 6 bp homology can significantly stabilize joint molecule formation (***Figure 4L***). Similarly, we fitted the data for $L_h$ = 7 nt using HMM analysis (191 molecules, 3819 transitions) and extracted the transition rates and number of transitions from the resulting TDP (***Figure 4K***). As expected, we observed a slower rate of leaving HS1 and HS2 for $L_h$ = 7 nt compared to $L_h$ = 6 nt (***Figure 4L***). Also, the dissociation rate of dsDNA bound to HS1 was higher than that of dsDNA bound to HS2 possibly because HS1 is located at the filament end where the terminal RecA monomer at the ssDNA–dsDNA junction binds with altered kinetics (***Joo et al., 2006***) as opposed to HS2 which is embedded within the RecA filament and benefits from complementary basepairing with the homology site and stabilizing interactions with neighboring RecA monomers.

Interestingly, transitions did not occur solely between neighboring sites (NH<->HS2 and HS1<->HS2 type transitions) but also involved transitions of the type NH->HS1 indicating that the RecA filament might overlook a region of homology during the sliding process. Homology recognition and base pairing of dsDNA at either HS1 or HS2 during sliding is a stochastic process. For a dsDNA initially bound to a non-homologous site (NH), the efficiency with which the first homology site the dsDNA encounters, HS2, is recognized before the more distant homology site, HS1, is recognized is given by,

$$E_{recognition}^{NH->HS2} = \frac{N^{NH->HS2}}{\left(N^{NH->HS2} + N^{NH->HS1}\right)},$$

where $N^{NH->HS1}$ and $N^{NH->HS2}$ denote the number of transitions from NH->HS1 and NH->HS2 respectively. For 6 nt homology ($L_h$ = 6 nt) the efficiency of homology site recognition, $E_{recognition}^{NH->HS2}$ is approximately 55%. This efficiency increased for 7 nt homology ($E_{recognition}^{NH->HS2}$ for $L_h$ = 7 nt is approximately 77%). Cumulatively, our data suggest that the efficiency of homology recognition by the RecA filament increases with the number of homologous nucleotides available for basepairing. We could not perform a similar analysis for $L_h$ = 8 nt since dsDNA bound to HS1 and HS2 positions in this case exhibited stable FRET states and transitions between them were rare (***Figure 4—figure supplement 2***).

## Discussion

Our study establishes RecA filament sliding as a possible mechanism to assist homology search during strand exchange reaction. Until now, there has been no example of a multi-protein complex bound to DNA such as the RecA filament which is capable of sliding along a second DNA strand to locate a matching sequence of bases. We estimated the diffusion constant, $D_{slide}$, for RecA filament sliding to be approximately $0.9 \times 10^{-3}$ µm²/s or 7700 bp²/s. Because the lifetime of heterologous synapses ranges from 0.5 to 10 s (*Figure 1—figure supplement 2B*) (*Mani et al., 2009*; *Forget and Kowalczykowski, 2012*), we estimate that dsDNA would diffuse over a length of 60–300 bp during each encounter with a RecA filament prior to its dissociation.

Using the estimated diffusion coefficient and sliding distance, we calculated the effective rate enhancement of target search due to sliding which is given by the ratio of the rate of association of RecA filament with the target site in the presence of sliding to the rate of association with the target site in the absence of sliding (D = 0) (*Hammar et al., 2012*). In this hypothetical model, RecA filaments can bind non-specifically to DNA but do not slide (*Figure 4—figure supplement 3*). Though, we do not know the exact microscopic rate constants which determine the binding probability of the RecA filament to the target site ($p_{bind}$), we can estimate that even for a recognition probability as low as approximately 1%, sliding leads to a rate enhancement in homology search by approximately 200-fold. If no sliding is allowed and homology must be recognized via fortuitous 3D contact that aligns the dsDNA and filament in perfect registry, homology recognition would be about two orders of magnitude slower. Therefore, 1D sliding combined with intersegmental transfer proposed by *Forget and Kowalcyzkowski (2012)* can dramatically accelerate homology search.

We cannot rule out the possibility that changes in angle between the dsDNA and the RecA filament (off-axis motions) might affect the observed FRET changes. Incorporating such effects of off-axis motions into the model would increase the time scale of linear sliding but the effect will be small because control experiments such as the DNA length dependence (*Figure 2C*) showed that on-axis motion is the dominant source of FRET fluctuations. Although our experimental scheme cannot probe for small and dynamic off-axis motions between the RecA filament and dsDNA, we can at least rule out the extreme possibility of 'dsDNA flipping' in which case the dsDNA may undergo a 180° change in orientation while remaining bound to the RecA filament (*Figure 4—figure supplement 4*). Therefore, a dsDNA that binds to a RecA filament in the wrong orientation cannot establish the correct orientation required for homology recognition and basepairing without full dissociation and rebinding.

We also demonstrated that homology recognition and base pairing processes can occur during sliding. Our results showed that as few as 6 nt of complementary base pairs are sufficient to act as the unit of homology recognition for RecA mediated homology search. It is noteworthy that recognition of a homology site is a stochastic process and the efficiency of homology recognition by the RecA filament depends on the length of homology which is encountered at a particular site. Furthermore, we show that adding even a single nucleotide to the minimum recognition unit of 6 nt improves recognition efficiency by approximately 1.5-fold.

Proteins mediate target search reactions by a combination of 1D sliding and 3D diffusion based processes (*Gorman and Greene, 2008*). While intersegmental transfer serves to bring non-contiguous segments of dsDNA close to each other, 1D sliding during homology search could help RecA reorganize the initial synaptic complex in *cis* without the need for a RecA filament to fully dissociate and rebind to a nearby homology site located within a short distance (up to few hundred base pairs). Given the ability of the RecA filament to slide between nearby homology sites (*Figure 4*), sliding could serve as a mechanism to rapidly scan neighboring sequences for the existence of an optimal seed sequence from which base pair propagation and heteroduplex extension reactions can proceed. An earlier report demonstrated sliding of Rad51 oligomers (not a Rad51 filament bound to ssDNA) without any clear functional role attributed to the observed sliding activity (*Graneli et al., 2006*). Because of significant structural and functional similarities between RecA and Rad51 (*Conway et al., 2004*), it is likely that sliding might also play a role in homology search mediated by Rad51.

## Materials and methods

### DNA preparation

DNA oligos used in our measurements were purchased from Integrated DNA Technologies (IDT, Coralville, IA). The oligos were suspended in T50 buffer (10 mM Tris-Cl, 50 mM NaCl, pH 8.0).

The DNA sequences used in our measurements are specified in *Supplementary file 1*. Double strand DNA was prepared by mixing complementary DNA molecules and heating to 90°C followed by slow cooling to room temperature over a period of 2 hr. dsDNA was purified from free ssDNA using a 12% native PAGE gel to ensure the absence of free ssDNA in our dsDNA preparations. The partial duplex DNA molecules for immobilization were prepared by annealing with the biotin_DNA sequence. All DNA molecules were labeled at the terminal ends (5' or 3') with Cy3 or Cy5 (labeling performed by IDT) as specified in the experimental scheme. For internally labeled DNA oligos, Cy3 N-hydroxysuccinimido (NHS) ester and Cy5 NHS ester (GE Healthcare) were internally labeled to a dT of ssDNA modified via a C6 amino linker (IDT, Coralville, IA).

## Single molecule RecA filament and DNA binding assay

The quartz (Finkenbeiner, Waltham, MA) surface is passivated with Polyethylene glycol (m-PEG-5000; Laysan Bio Inc.) and 1–2% biotinylated PEG (biotin-PEG-5000; Laysan Bio Inc., Arab, AL). The coating of the quartz imaging surface with PEG (*Roy et al., 2008*) eliminated effects due to non-specific binding of proteins. Acceptor labeled reference ssDNA molecules were immobilized on the passivated surface by means of a biotin–neutravidin interaction. After washing away excess of acceptor molecules, the reference ssDNA was incubated with 1 μM RecA (Epicenter biotechnologies, Madison, WI) and 1 mM ATPγS (EMD Calbiochem, Billerica, MA) in an incubation buffer containing 25 mM Tris Acetate pH 7.5, 100 mM Sodium Acetate and 1 mM Magnesium Acetate. In some cases, 1 mM ATP was used instead of ATPγS. RecA ΔC17 protein was a generous gift from Dr. M. M. Cox (University of Wisconsin). After incubation for 15 min to ensure complete filament formation on ssDNA ($L_{filament}$) molecules, the buffer in the chamber was exchanged with a solution of non-homologous dsDNA (1 nM) and 1 mM ATPγS in a strand exchange buffer (25 mM Tris Acetate pH 7.5, 100 mM Sodium Acetate, 10 mM or 1 mM Magnesium Acetate) supplemented with an oxygen scavenging system (1 mg/mL glucose oxidase, 0.8% glucose, 0.04 mg/mL catalase and 3 mM Trolox). Imaging was initiated as soon as the buffer exchange was complete. All measurements were carried out at room temperature (23 ± 1°C).

## Single molecule data acquisition

Excitation of the donor, Cy3, was carried out using a Nd:YAG laser (532 nm, 75 mW; Crystalaser, Reno, NV) by means of prism type total internal reflection microscopy (*Roy et al., 2008*). After filtering the scattered excitation light using a 550 nm long pass filter, fluorescence emission from the donor and the acceptor was refocused onto an EMCCD camera (Andor, UK). The Cy3 and Cy5 emissions were split into two channels using a 630 nm dichroic mirror. In the case of three color measurements involving Cy3, Cy5 and Cy7, an additional dichroic mirror (730 nm) was used to split the emission across three emission channels. The time resolution for all single molecule strand exchange experiments was 30 ms unless otherwise specified. The data acquisition was carried out using home built software written in Visual C++. The movies obtained with the CCD were analyzed first using IDL and the intensities of the fluorophores and the time traces was visualized using customized MATLAB programs (*Joo and Ha, 2008*; *Roy et al., 2008*).

## Single molecule data analysis

The dwell time analysis was carried out by a home-written MATLAB program. The background intensity in the donor and acceptor channel was subtracted followed by leakage subtraction of the donor signal to the acceptor channel. Details regarding the acquisition and analysis are based on previously published methods. After visually inspecting the acquired data, we manually selected the relevant time periods for analysis and used Origin 8.0 to plot the data. Cross-correlation analysis was performed as previously described (*Kim et al., 2002*) by calculating cross-correlation functions for donor and acceptor time traces for a given molecule. Fitting the averaged cross-correlation functions to a single exponential function allows to estimate the average cross correlation time for each measurement. Three color data analysis was performed as previously described (*Hohng et al., 2004*; *Roy et al., 2009*). Following leakage correction for Cy3, Cy5 and Cy7 intensities, we additionally corrected Cy7 intensity by utilizing a gamma factor (to correct for differences in optics and detection efficiency).

$$\gamma = \frac{\Delta I_{Cy5}}{\Delta I_{Cy7}}$$

$\Delta I_{Cy5}$ and $\Delta I_{Cy7}$ represent the ratio of the change in intensity in Cy5 and Cy7 channels which is calculated by following the change in intensity in the two channels following Cy7 photobleaching. $E_{Cy3->Cy5}$,

the FRET efficiency corresponding to Cy3–Cy5 interaction and $E_{Cy3->Cy7}$ corresponding to the Cy3-Cy7 interaction was calculated as

$$E_{D-A,i} = \frac{I_{A,i}}{\Sigma I_{A,i} + I_D},$$

where $I_D$ and $I_A$ are the donor and acceptor intensities respectively. The suffix, i, denotes the two acceptors Cy5 and Cy7 utilized in this measurement. We carefully designed the three color measurement to ensure that the two acceptors Cy5 and Cy7 spaced far apart from each other. such that $E_{Cy5->Cy7} \sim 0$ and the presence of a stable RecA filament (by using ATPγS) results in the spacing between Cy5 and Cy7 remaining unchanged during the course of the measurement. Software for acquiring and analyzing single molecule FRET data is freely available for download from https://physics.illinois.edu/cplc/software.

## Monte Carlo simulation for calibration

Monte Carlo (MC) simulation of the free diffusion along dsDNA was written in MATLAB. Since the persistence length of RecA-ssDNA was reported to be approximately 784 nm (approximately 2300 bp) (*Hegner et al., 1999*), the RecA-DNA filaments in our experiments can be simply treated as a rod in one dimension. Equal probabilities of moving forward or backward were generated by the program. The step size of the MC simulation was set to be small enough (0.1 or 0.01 bp, in our case) to mimic continuous sliding of the RecA filament along dsDNA. The distance between the donor and acceptor fluorophores and their corresponding FRET value was calculated after each step. Since RecA binding to ssDNA results in an average rise between successive nucleotides of d = 5.1 Å/bp, L could be expressed by

$$L = \sqrt{r^2 + \left(d \times |x - x_o|\right)^2},$$

where $x$ and $x_o$ respectively represent the position of non-homologous dsDNA and the static acceptor along the dsDNA. Total intensity of each data bin, 30 ms, is set to be 500 a.u., matching our typical data intensity in TIRF experiments. Gaussian white noise with a peak intensity of 40 a.u. is applied. Finally, same cross correlation (CC) analysis as described before was applied to the MC simulated traces for calibration purpose (τ, dwell time of CC analysis; D, diffusion coefficient).

## Acknowledgements

We dedicate this paper to the memory of our brilliant colleague, Robert MClegg (1945-2012), a pioneer in the application of fluorescence and FRET approaches to the life sciences, a great mentor and a remarkable human being. We thank K Lee, J Yoo and C Joo for discussions and experimental help. We are grateful to A Jain for help with data acquisition. We thank A Jain and R Zhou for careful reading of the manuscript. We also thank A Jain, J Fei, H Koh, Y Ishitsuka, S Doganay and X Shi for advice and comments related to the project.

## Additional information

### Funding

| Funder | Grant reference number | Author |
|---|---|---|
| National Science Foundation | PHY-0646550 | Taekjip Ha |
| National Science Foundation | PHY-0822613 | Taekjip Ha |
| National Institutes of Health | GM065367 | Taekjip Ha |
| Howard Hughes Medical Institute | | Taekjip Ha |

The funders had no role in study design, data collection and interpretation, or the decision to submit the work for publication.

## Author contributions

KR, Conception and design, Acquisition of data, Analysis and interpretation of data, Drafting or revising the article; CL, Analysis and interpretation of data, Contributed unpublished essential data or reagents; TH, Conception and design, Analysis and interpretation of data, Drafting or revising the article

## Additional files

### Supplementary files

- Supplementary file 1. DNA sequences used in measurements

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
