## [Author Response]

*1) All data interpretation and analysis are based on the notion that the RecA-ssDNA filament and the duplex DNA act as two stiff, coaxial rods, and that all relative motion involves linear displacement (sliding) of these two rods relative to one another. This interpretation ignores the possibility of off-axis motion (swinging or other fluctuations), which may be relevant to homology search. While the reviewers find the evidence for sliding, especially data shown in Figures 2 and 3, strong, they think that potential contributions from the off-axis motion, and how they may affect data interpretations and simulations, should be discussed. The authors should discuss this effect in the revised manuscript and, if necessary, provide additional data or analysis to support their arguments. The assay shown in Figure 4 may help assess off-axis motion. In this assay, one expects ∼50% of the binding events to occur in the correct orientation that permits recognition of HS1/HS2, and the remaining 50% to occur with the duplex DNA in the incorrect orientation, which may not lead to recognition of HS1/HS2. A statistical analysis of the fractions may provide clues for whether reorientation or off-axis motion between the duplex DNA and RecA-ssDNA filament occurs. The authors may also find a modification of the Figure 4 assay helpful, namely change the sequence of the HS2 site, such that it would be complementary to a sequence on the opposing strand of the incoming duplex, and then test whether recognition still occurs for both HS1 and HS2 sites or only one of them in a single binding event*.

This is a good point. The angle between dsDNA and the RecA filament may indeed change during the sliding process and we may call such angle changes as ‘off-axis motion’. We indeed cannot rule out such off-axis motion as one of the sources for the observed FRET fluctuations but our controls including the DNA length dependence (Figure 2) suggest that the ‘on-axis motion’ is the dominant source. Nevertheless, contributions from any ‘off-axis motion’, if significant, could affect the apparent diffusion coefficient that we estimate based on our modeling that considers only the on-axis motion. We added the following sentence to the Discussion to acknowledge the possibility of ‘off-axis motion’:

“We cannot rule out the possibility that changes in angle between the dsDNA and the RecA filament (off-axis motions) might affect the observed FRET changes. Incorporating such effects of off-axis motions into the model would increase the time scale of linear sliding, but the effect will be small because control experiments such as the DNA length dependence (Figure 2C) showed that on-axis motion is the dominant source of FRET fluctuations.”

We wanted to address the possibility of an extreme ‘off-axis motion’ where the dsDNA may undergo a 180 degrees change in orientation, that is, the dsDNA may flip its orientation during homology search. Although this is very unlikely *in vivo* considering the presence of kilobase long filaments and DNA in the cell, previous single molecule FRET measurements reported the flipping of protein orientation on DNA without full dissociation (Abbondanzieri et al., 2008). Therefore, we performed control experiments to test the possibility that short dsDNA may flip its orientation while remaining bound to the RecA filament. We used the ssDNA substrate that contains two 8 nt homology sites embedded in poly(dT) – the same DNA we used to demonstrate the existence of two stable positions that are interchanging only infrequently (Figure 4–Figure Supplement 2). If you use a dsDNA with the 8 nt homology but with the donor attached at the opposite end of the duplex DNA, we observe rapid fluctuations over a broad range of FRET values (new Figure 4–Figure Supplement 4A, B), identical to what’s observed if the ssDNA is made only of poly(dT) (new Figure 4–Figure Supplement 4C, D). The rapid fluctuations indicate the lack of homology recognition. This is likely because when the dsDNA binds to the RecA filament with the wrong orientation, it cannot flip its orientation to find the homology sites (HS1 and HS2). If flipping can occur while the dsDNA is bound to the RecA filament we would have observed stable low FRET states that can transit to the rapidly fluctuating phase and back. For this construct, when the dsDNA binds to the filament in the correct orientation, where the unlabeled homologous end is in close proximity to the acceptor, FRET stays low and these events were not included in the histogram shown in Figure 4–Figure Supplement 4B. We have revised the manuscript to formally discuss the possibility of flipping in the Discussion section:

“Although our experimental scheme cannot probe for small and dynamic off-axis motions between the RecA filament and dsDNA, we can at least rule out the extreme possibility of ‘dsDNA flipping’ in which case the dsDNA may undergo a 180 degree change in orientation while remaining bound to the RecA filament (Figure 4–Figure Supplement 4).”

*2. The authors suggest that the transitions between the HS1 and HS2 sites observed in Figure 4 are due to sliding. Indeed, the scenario that the unwound ssDNA first detach from the HS1 site, re-anneal with the complementary strand, slide along the RecA filament and then bind to the HS2 site is a highly likely one. However, give the relatively short distance between the HS1 and HS2 sites (5 nucleotides in between), is it possible that the ssDNA segment can transition between the two homology sites through a different mechanism? The authors should discuss whether the 5-nucleotide distance is sufficiently long to conclude that transitions between the two sites are due to sliding, and if necessary, provide additional evidence*.

Although the gap between the homologous sites is 5 nt, the distance that the dsDNA has to transverse between the two sites is (5+L_h_) nt where L_h_ is the size of the homology. Therefore, even for the smallest Lh of 6 nt that gave homology recognition, the distance that the dsDNA has to travel is 11 nt or about 50 angstroms (11*3.4*1.5) because RecA filaments are stretched by about 1.5 fold compared to B-form DNA. We cannot think of mechanisms other than sliding that can cause rapid movement over such length scales. We added a sentence to the Results section to address this point:

“Although the gap between HS1 and HS2 is 5 nt, it is noteworthy that the total distance traversed by dsDNA to exhibit complete basepairing is (L_h_+5)nt. Thus, for the smallest L_h_ of 6 nt, this translates to a distance of ∼ 50 Ǻ (11nt * 3.4 Ǻ * 1.5). Given the rapid movements of the dsDNA between adjacent homology sites and the fact that both the dsDNA and RecA filament are stiff at the length scales of our experiments, we cannot think of mechanisms other than sliding to explain the observed transitions.”